# Effect of testing criteria for infectious disease surveillance: The case of COVID-19 in Norway

**Solveig Engebretsen**👤*, **Magne Aldrin**

SAMBA, Norwegian Computing Center, Oslo, Norway

* solveig.engebretsen@nr.no

## Abstract

During the COVID-19 pandemic in Norway, the testing criteria and capacity changed numerous times. In this study, we aim to assess consequences of changes in testing criteria for infectious disease surveillance. We plotted the proportion of positive PCR tests and the total number of PCR tests for different periods of the pandemic in Norway. We fitted regression models for the total number of PCR tests and the probability of positive PCR tests, with time and weekday as explanatory variables. The regression analysis focuses on the time period until 2021, i.e. before Norway started vaccination. There were clear changes in testing criteria and capacity over time. In particular, there was a marked difference in the testing regime before and after the introduction of self-testing, with a drastic increase in the proportion of positive PCR tests after the introduction of self-tests. The probability of a PCR test being positive was higher for weekends and public holidays than for Mondays-Fridays. The probability for a positive PCR test was lowest on Mondays. This implies that there were different testing criteria and/or different test-seeking behaviour on different weekdays. Though the probability of testing positive clearly changed over time, we cannot in general conclude that this occurred as a direct consequence of changes in testing policies. It is natural for the testing criteria to change during a pandemic. Though smaller changes in testing criteria do not seem to have large, abrupt consequences for the disease surveillance, larger changes like the introduction and massive use of self-tests makes the test data less useful for surveillance.

**Data Availability Statement:** All files are available from the Norwegian Institute of Public Health at https://github.com/folkehelseinstituttet/surveillance_data (Version from November 15, 2022).

## Introduction

The first confirmed case of SARS-CoV-2 in Norway was confirmed on February 26, 2020 [1], two weeks before the World Health Organization declared the outbreak as a pandemic on March 3, 2020 [2]. As a response, the Norwegian government implemented a nationwide lockdown on March 12, 2020, including closing of kindergartens and schools [3].

Confirmation of infection by testing has two main purposes: 1) finding out who is infectious in order to prevent spreading of the disease, and 2) surveillance, which is the focus of this study. There are many alternative data sources which can be used for surveillance of infectious diseases. The different surveillance data sources have different advantages and limitations, in

**Funding:** SE and MA were funded by the Research Council of Norway (basic funding, https://www.forskningsradet.no/en/). The funders had no role in study design, data collection and analysis, decision to publish, or preparation of the manuscript.

**Competing interests:** The authors have declared that no competing interests exist.

terms of for example accuracy, representativity, noise, timeliness, acceptance, and cost and resource demands [4,5]. It is therefore common to combine evidence from multiple surveillance data sources. Specifically, combining evidence in different data sources was also recommended by WHO for COVID-19 surveillance purposes [6]. Another example is the routine influenza surveillance system in Norway which includes, among other sources, data on the number of general practitioner visits related to influenza, sentinel surveillance and laboratory-based surveillance of patients with acute respiratory infection, seroprevalence studies, and hospital prevalence [7]. Laboratory-based surveillance and seroprevalence studies are in general less noisy than data on influenza-like illness diagnoses from general practitioners, but more delayed and costly.

During the COVID-19 pandemic, data on confirmed cases were used for surveillance and modelling purposes in many countries, for example when estimating reproduction numbers (see for example [8–11]). Specifically, confirmed cases were included in the WHO's global COVID-19 surveillance system, in addition to data on e.g. the total number tested, deaths and hospitalisations due to COVID-19 [12]. Moreover, WHO's guidance for public health surveillance of COVID-19 states that data on number of confirmed cases from laboratory tests should be collected as they are useful for early warning of changes in epidemiological patterns and to monitor trends in infection [6]. During the COVID-19 pandemic, multiple countries also used environmental surveillance, in particular wastewater surveillance. Such environmental surveillance has the advantage of not being affected by health care seeking behaviour, disease severity, or testing criteria/capacity [13]. Norway also used wastewater surveillance from June 2022 [10]. A systematic review on the use of environmental surveillance during COVID-19 concludes that wastewater surveillance could be used as an early warning signal and hence complement clinical surveillance of SARS-CoV-2 [14], but that the cost-benefit needs to be taken into account. Wastewater surveillance could also contribute to early detection of new variants. As noted by WHO, laboratory-confirmed cases remain the primary source of evidence on incidence of infection [13].

The Norwegian Institute of Public Health used both hospital incidence and confirmed cases for inference in the mathematical models of COVID-19 spread and for estimating reproduction numbers [15,16]. Hospital incidence is a more robust surveillance source than test data, as it does not depend on testing criteria and testing capacity. Hospital incidence is also less sensitive to health seeking behaviour and self-selection. In addition, one of the main uses of mathematical modelling for real-time situational awareness is to predict hospitalisations, and hence it is important to calibrate to hospital data. There are, however, fewer observations in the hospital data than in the test data, they are more delayed, and they generally represent a population at higher risk than the community at large. Moreover, it is problematic to use hospital case data for surveillance if the hospital capacity is reached. To our knowledge, this was not a problem during the pandemic in Norway. Hence, one of the largest limitations with using hospital incidence data alone in infectious disease surveillance is that it takes more time before a change in transmission is visible in the hospital data than in the test data. It is therefore important to combine information from both data sources when calibrating mathematical disease spread models and in surveillance. Though data on confirmed cases from laboratory tests are subject to under reporting due to e.g. health care seeking behaviour, they can still be used relatively straightforward to monitor trends as long as the under reporting stays constant [5]. Nonetheless, it is difficult to use test data in periods where the testing strategy is unstable, that is, when the meaning of a positive test changes over time. If the meaning of a positive test changes, this will potentially have consequences for surveillance and estimated reproduction numbers [17]. Changes in testing criteria may lead to conflicting information in different data sources used for surveillance and/or for inference in mathematical models. If the changes in

testing criteria have a practically significant effect on the outcomes of the models, then a complex observation model for the test data would be necessary. This is a nontrivial inference problem and will likely be subject to identifiability issues in a complex observation model unless the parameters are assumed fixed or given informative priors. In Norway, the modelling team from the Norwegian Institute of Public Health handled this in part by incorporating the total number of tests in a relatively simple observation model, as a proxy for changes in testing criteria and testing capacity. The test data were not used for model inference in the early period when testing had not stabilised. Moreover, a seven days rolling mean of the test data was used in the calibration, in order to take into account potential day-of-the-week effects [15]. Using the number of tests to estimate the reproduction number for COVID-19 was also done in for example [18,19], and another approach to an observation model which takes into account changes in testing criteria can be found in [20].

The testing criteria and testing capacity changed numerous times during the COVID-19 pandemic in Norway. Changes in testing strategy may affect the relationship between the number of positive PCR tests and the total number of PCR tests. Interventions and restrictions on social contacts (and hence also liftings/easings) may also affect the test data, for example by making contact tracing easier such that close contacts may be traced and tested.

In conjunction with the nationwide lockdown on March 12, 2020, there were also changes in the testing criteria. The effects of these changes have previously been studied [1]. The testing criteria changed from prioritising close contacts of confirmed cases and individuals who had travelled to countries with early-confirmed cases to prioritising symptomatic individuals with comparatively high risk of severe disease and health personnel. This led to a change in the demography of the population of positive test cases [1], which is hard to tease out from the changes in the society at large due to the social contact restrictions. In this study, we only have access to PCR test data from April 1, 2020.

There have been multiple changes in testing criteria and capacity also after March 12, 2020 [21]. Some of the most important changes in testing criteria and restrictions were:

- On April 29, 2020, the testing capacity had increased substantially compared to early in the pandemic, and thus all suspected cases of COVID-19 by general practitioners were recommended to get tested [21].

- From August 12, 2020, a referral from a general practitioner was no longer necessary for testing [21].

- Restrictions on social contacts were implemented on October 26, 2020, with the aim of reducing spread of infection and easing the contact tracing [22], with additional reinforcements on November 5, 2020 [23].

- Mandatory border testing was implemented on January 2, 2021 when travelling to Norway from certain countries with high rates of infection [24].

- Restriction on social contacts was implemented on March 25, 2021 [25].

- A gradual reopening was carried out on April 16, 2021 [26].

- Further reopening was carried out on May 27, 2021. At the same time, Norway opened the possibility for testing without suspicion of COVID-19 infection and mass testing [27]. Mass testing was used for example in conjunction with social events around term start in August, 2021 at the universities [28].

- On September 25, 2021 the country was fully reopened [29] and the contact tracing was drastically downgraded [30].

- Due to the omicron variant, new restrictions were implemented in December 2021, along with a recommendation of regular testing in schools [31–33].

- Corresponding easing of restrictions was gradually implemented on January 14, 2022 [34], February 1, 2022 [35], and February 12, 2022 [36]. From January 14, 2022, the contact tracing was further reduced and there was an increased use of self-tests at home [37]. On January 28, 2022, regular testing was no longer recommended by the government [38] and all contact tracing ended on February 12, 2022 [39].

- On March 28, 2022, the mobile test services in the capital Oslo were closed [40].

The first self-tests became available for purchase over-the-counter in grocery stores and pharmacies in Norway during June/July 2021 [41,42]. The testing criteria thus changed drastically during summer and autumn of 2021, with increased use of mass testing and access to self-tests. Confirmation of positive self-tests with PCR tests was recommended [43]. Hence there was a change in the population of individuals testing themselves with PCR tests towards mainly including individuals who had already tested positively on a self-test at home. There was no systematic reporting and registration of self-tests in Norway.

In this paper, we wish to study which factors affect the total number of tests and the relationship between the number of positive and the total number of PCR tests. Specifically, we investigate the effect of weekday, public holiday and changes in testing criteria and restrictions, including the use of self-tests at home.

## Materials and methods

### Data

The data are publicly available online [44]. We used the newest versions of the hospital incidence data and test data, dating from November 15, 2022. The data were accessed and downloaded on August 3, 2023. We did not have access to any information that could identify individuals in the data. The test data contain the daily number of negative and positive COVID-19 PCR tests between April 1, 2020 and November 13, 2022. The only date missing in the data is May 30, 2020. Hence, we do not have access to test data from the very start of the pandemic. The hospital data contain daily counts of individuals admitted to hospital with COVID-19 as main cause (i.e. daily hospital incidence). These data were publicly available and reported weekly by the Norwegian Institute of Public Health during the pandemic. The Norwegian Institute of Public Health's web page with test criteria in Norway [21] has been archived but was accessed through the Wayback Machine [45].

### Methods

We analysed the data using two different approaches. The first was a purely explorative data analysis, where we inspected the data from the whole period available, that is, from April 1, 2020 to November 13, 2022. In addition, we fitted regression models to the daily test data from April 1, 2020 to January 1, 2021 (in total 274 days, one day missing). We chose to limit the regression analysis to this period only, as it was not cluttered by larger known changes in variant and by vaccination.

**Regression models.** We fitted three generalised additive regression models (GAM) [46] to the test data for 2020, with flexible, smooth effects of the continuous explanatory variables (i.e. smoothing splines). All analysis was performed in R 4.2.3 [47].

In Model 1 we fitted a GAM for the total number of tests, $Y$, assuming a gamma distribution with a logarithmic link function, so that

$$Y_d \sim \text{Gamma}(\mu_d, v),$$

where $Y_d$ is the number of tests for day $d$, $\mu_d$ is the expectation and $v$ is a dispersion parameter. As explanatory variables we included smoothing splines in hospital incidence and time, and parametric effects for type of weekday. We used four different levels for weekday: Monday, Tuesday-Friday (reference level), Saturday-Sunday, and public holiday. The expectation was modelled as

$$\log(\mu_d) = \beta_0 + s(h_d) + s(d) + \beta_1 x_{1,d} + \beta_2 x_{2,d} + \beta_3 x_{3,d},$$

where $h_d$ is the hospital incidence on day $d$, $d$ denotes day, $s$ denotes a smoothing spline, $x_{d,1}$ $x_{d,2}$, $x_{d,3}$ are indicator variables for whether day $d$ was a Monday, Saturday-Sunday or public holiday, respectively, and $\beta_0$, $\beta_1$, $\beta_2$, and $\beta_3$ are regression coefficients which we estimated. Assuming a multiplicative model (implied by a logarithmic link function) is reasonable as the total number of tests changed substantially over the study period.

Model 2 is a logistic GAM for the probability of a positive PCR tests, with the same explanatory variables as for Model 1, such that

$$Y_d \sim \text{Bin}(n_d, p_d),$$

$$\text{logit}(p_d) = \beta_0 + s(h_d) + s(d) + \beta_1 x_{1,d} + \beta_2 x_{2,d} + \beta_3 x_{3,d},$$

where $Y_d$ is now the number of positive tests for day $d$ and assumed to be binomially distributed with success probability $p_d$ and number of trials $n_d$ as the total number of tests on day $d$. We fitted an alternative model in addition, Model 3. In this model we included smoothing splines in the total number of tests and hospital incidence as explanatory variables, such that

$$\text{logit}(p_d) = \beta_0 + s(T_d) + s(h_d),$$

where $T_d$ is the total number of tests on day $d$. We fitted two alternative models for the probability of a positive PCR test as the different variables were correlated, and hence it is difficult to simultaneously adjust for e.g. total number of tests and weekday. The models are approximately multiplicative since the probabilities are small, and the odds ratios are approximately equal to the relative risks for low probabilities. We thus obtain approximate relative risks by this assumption.

We adjusted for hospital incidence as a proxy for the infection level in the population. The test data could not be used as a proxy for the infection level, since they are the response variable of the study. Because the time from transmission to hospital admission is longer than the time to positive test, these two data sources will be shifted in time. We chose to use a time shift of seven days, as this gave the largest autocorrelation (0.83) between these two time series. Moreover, we used a seven-days rolling mean of the hospital incidence, to account for potential day-of-the-week effects.

In the figures showing the effects of the different variables, the effects are scaled such that the average of the effects equals the average of the corresponding response variable in the data.

Note that we also tried out models which allowed for sudden jumps in accordance with the changes in testing criteria/policy. However, it turned out that the changes in number of tests and positive tests in 2020 were gradual and did not occur as sudden jumps.

**Uncertainty.** Since the observations are dependent in time and there is potential overdispersion, we computed uncertainty estimates through a circular block-bootstrap [48], as

explained in S1 Text. Bootstrap estimates were used as the standard approximated confidence intervals are too narrow with dependent and/or overdispersed observations.

## Results

Fig 1 shows the proportion of positive tests versus total number of tests with different colours for different time periods. The upper panel shows the whole time period. There was a distinct difference between the data before and after the massive use of self-tests (i.e. comparing before and after September 25, 2021). The different time periods after September 25, 2021 were also clearly separated. The proportion of positive tests was higher after September 25, 2021. The proportion of positive tests increased over time until the closing of mobile test stations on March 28, 2022.

The middle panel shows the data until 2021, that is, before vaccines and the alpha variant. There were distinct differences between the different time periods. In the start, there were relatively few tests and a high proportion of positive tests. Between April 29, 2020 and August 12, 2020 when everyone with suspected COVID-19 by a medical doctor were prioritised for testing, the proportion of positive tests decreased while the total number of tests increased. After August 12, 2020, testing was recommended for every suspected COVID-19 case. The number of tests increased, while the proportion of positive tests did not change. After restrictions were implemented on October 26, 2020, both the number and proportion of positive tests increased.

The data from January 1, 2021 until reopening on September 25, 2021 are plotted in the bottom panel. The periods were less clearly distinguishable. On May 27, 2021 Norway opened up to testing without suspicion of infection and mass testing. Not surprisingly, the proportion of positive tests seemed to decrease. More surprisingly, the total number of tests also seemed to decrease, however note that this is purely an exploratory analysis, and we do not adjust for e.g. the decreased infection level in summer. After the gradual increase in use of self-tests, there was a marked increase in the proportion of positive tests, but also the total number of tests. In the figure, August 12, 2021 marks the use of self-tests during social events at the universities in conjunction with term start.

### Regression models

**Total number tested.**   The estimated parametric effects of Model 1 are provided in Table 1. More tests were performed on Mondays (24%) and fewer tests were performed during weekends (64% less) than on Tuesdays-Fridays. Fewest tests were performed on public holidays (69% less). We did not find any effect of hospital incidence on the total number tested (Fig 2). The effect of time (Fig 2) showed two main regimes with different test capacities after adjusting for hospital incidence, that is, relatively few tests until August, 2020, a rapid increase in test capacity, and then a stable, relatively high level for the rest of 2020. Note that it is not possible to separate changes in testing capacity and changes in test-seeking behaviour due to changes in test criteria.

### Probability of positive test

The estimated parametric effects for Model 2 (odds ratios, approximately equal to relative risks) are provided in Table 1. There was a higher probability of testing positive on public holidays (ca. 70%) and weekends (ca. 37%) than for Tuesday-Fridays. The probability of testing positive was lowest on Mondays (ca. 8% lower than for Tuesday-Fridays). The results imply different testing criteria or test-seeking behaviour for different weekdays. Note that the

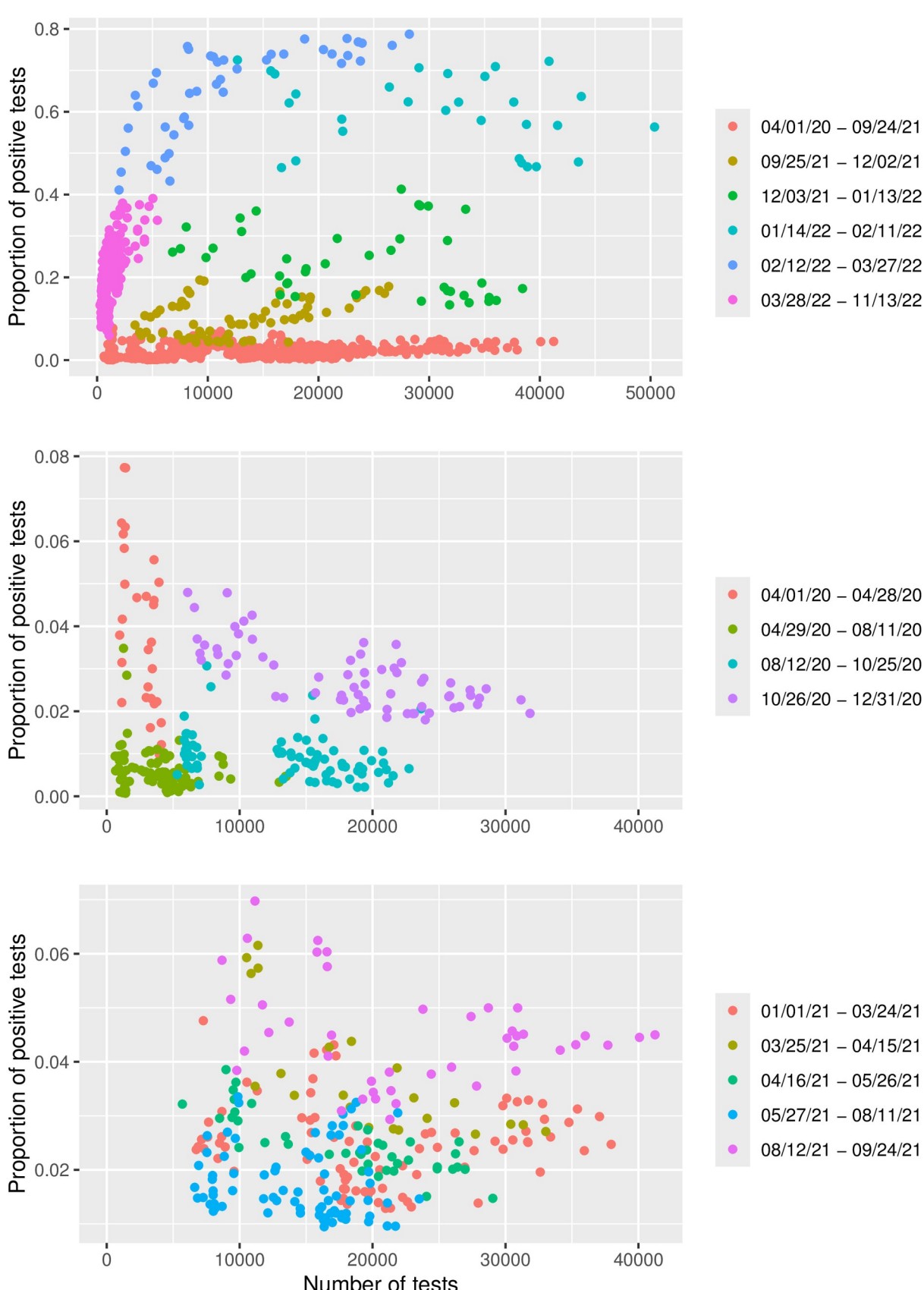

**Fig 1. Proportion of positive tests versus total number of tests.** The proportion of positive tests versus total number of tests where different time periods are marked with different colours.

estimated uncertainty for public holiday was relatively high, and that there were only 14 such days in the data.

The estimated effects of time and hospital incidence are provided in Fig 3. The probability of testing positive clearly changed over time, but it was unclear whether the changes were due to changes in test criteria. There was a slight, insignificant increase in the probability of testing positive with increased hospital incidence.

The estimated effects of the total number tested and hospital incidence for Model 3 are provided in Fig 4. The probability of testing positive decreased non-linearly with the total number of tests and flattened out. The effect of hospital incidence was approximately linear, that is, a different effect than for Model 2. This was likely because the different explanatory variables were correlated. Generally, both hospital incidence and total number of tests increased over time. If the test criteria, capacity, and hospital risk are constant, then we expect a linear relationship between the probability of testing positive and hospital incidence (with a corresponding time shift).

## Discussion

We found that both the proportion of positive tests and total number of tests changed during the pandemic, also after we adjusted for the hospital incidence. It was unclear whether this was as a direct consequence of the changes in test criteria and/or social restrictions. When the total number of tests increased, it is not possible to know whether this was due to increased test capacity, or because more people wanted to get tested, or a combination. In general, the proportion of positive tests increased over time, implying that the contact tracing may have improved over time. Specifically, the proportion of positive tests increased after introduction of restrictions on social contacts, which could imply more efficient/easier contact tracing with fewer social contacts. The test data clearly changed after the introduction of self-tests. The proportion of positive tests increased drastically. Moreover, the probability of a positive test increased with the total number of tests, and hence it is hard to use these data for disease surveillance.

On holidays and weekends, there were fewer tests and a higher probability of a positive test than for Tuesdays-Fridays. More tests and a lower probability of a positive test were found on Mondays. In general, the probability of testing positive decreased non-linearly with the total number of tests. This coincides with an analysis of COVID-19 test data from Ontario, Canada

**Table 1. Parameter estimates.**

| Model 1 | | | |
|---|---|---|---|
| | Monday ($\exp(\beta_1)$) | Saturday/Sunday ($\exp(\beta_2)$) | Public holiday ($\exp(\beta_3)$) |
| Exponentiated parameter estimate | 1.24 | 0.36 | 0.31 |
| 95% CI | (1.19–1.29) | (0.33–0.39) | (0.23–0.39) |
| **Model 2** | | | |
| | Monday ($\beta_1$) | Saturday/Sunday ($\beta_2$) | Public holiday ($\beta_3$) |
| Exponentiated parameter estimate | 0.92 | 1.37 | 1.7 |
| 95% CI | (0.86–0.97) | (1.29–1.45) | (1.35–2.18) |

Estimated relative risks for Model 1 and Model 2 (approximated) with corresponding 95% confidence intervals (CI).

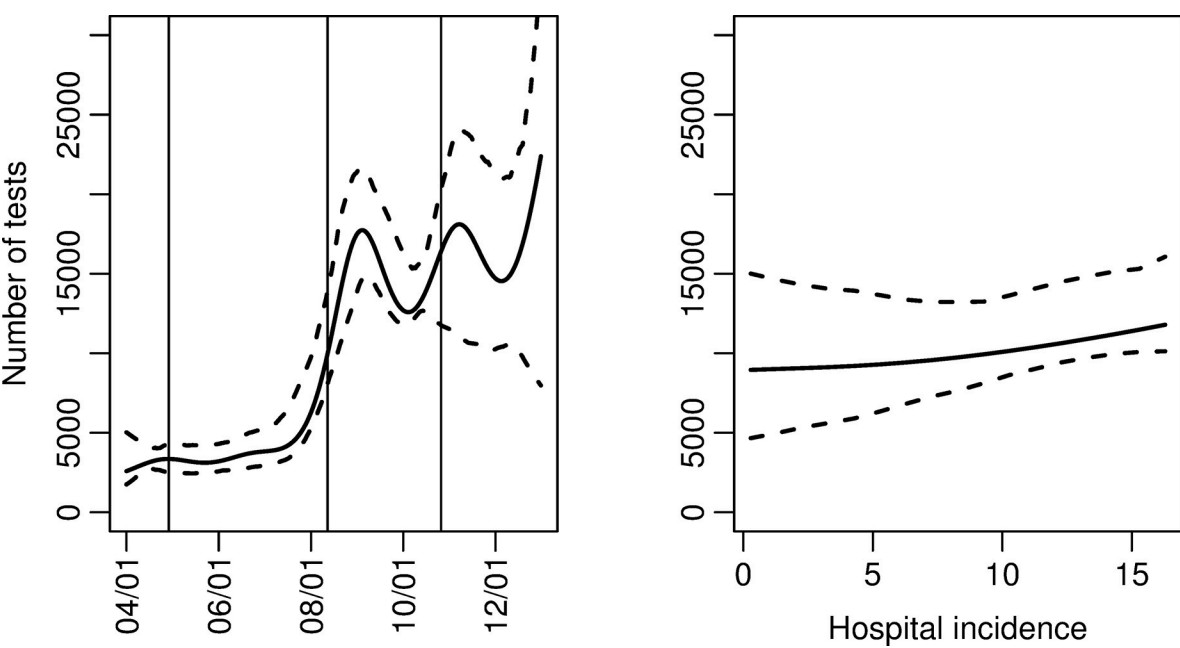

**Fig 2. Total number tested in Model 1.** Estimated effects of time and hospital incidence on total number tested with corresponding 95% confidence intervals. Changes in test criteria are marked with vertical lines for April 29, 2020, August 12, 2020, and October 26, 2020.

[49]. Different testing criteria and total number of tests for holidays means that it is more difficult to detect changes in transmission during holiday periods like Christmas and Easter. Correcting for the lower number of tests during holidays is a non-trivial task, due to the non-

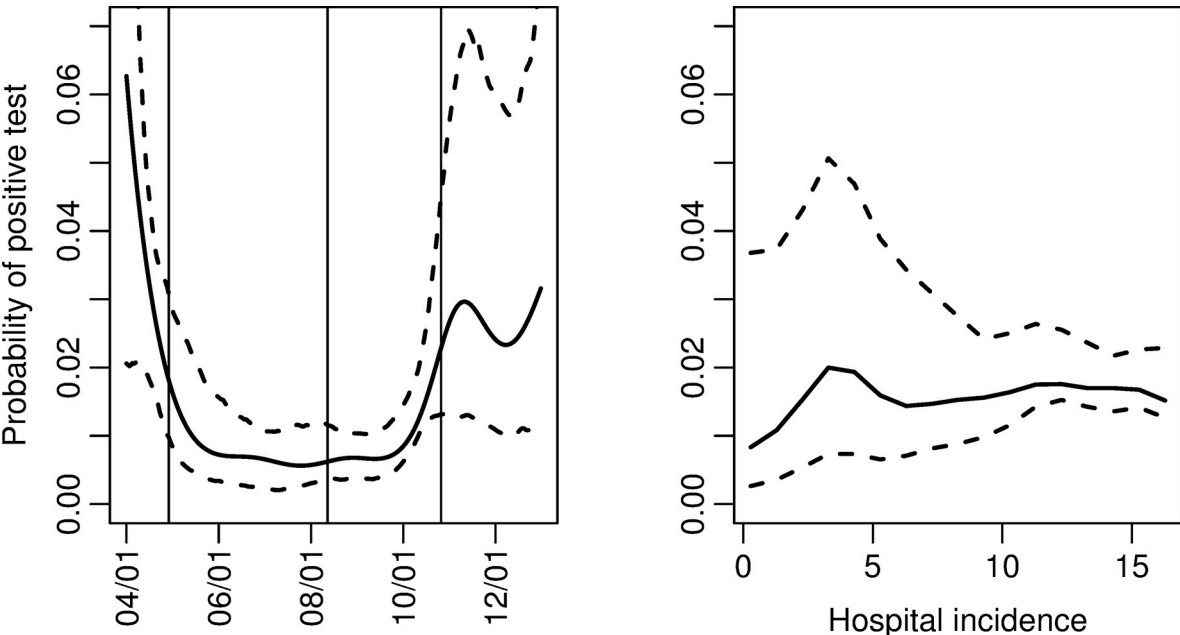

**Fig 3. Probability of testing positive in Model 2.** Estimated effects of time and hospital incidence on the probability of testing positive with corresponding 95% confidence intervals. The changes in test criteria are marked with vertical lines for April 29, 2020, August 12, 2020, and October 26, 2020.

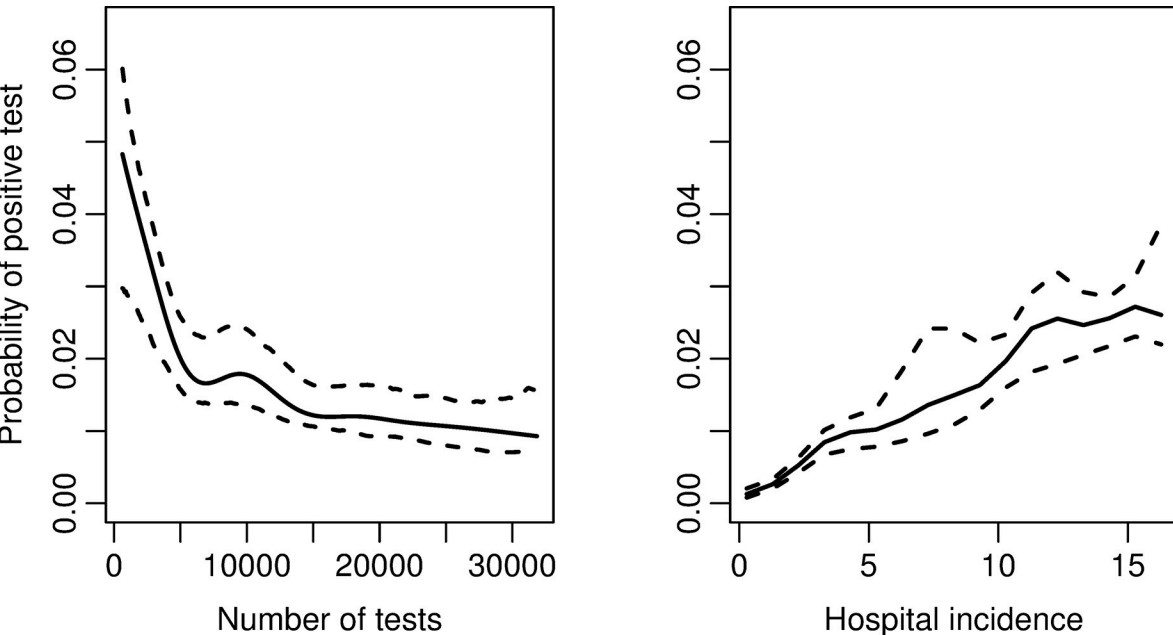

**Fig 4. Probability of testing positive in Model 3.** Estimated effects of total number tested and hospital incidence on the probability of testing positive with corresponding 95% confidence intervals.

linear/non-constant relationship between the probability of testing positive and the total number of tests.

We adjusted for the hospital incidence as a proxy for the infection level in the population. This is unproblematic as long as the hospital incidence is a constant proportion of the number of infected individuals. However, this would not be the case if there were changes in viral variants/properties such that the severity of infected cases changed. If the hospital capacity is reached, then the hospital incidence data need to be treated as censored data, and that would have to be taken into account in the analysis. To our knowledge, the hospital capacity was not reached in Norway during the pandemic.

Norway has a high general trust in the health authorities, partly due to the universal health care. A high general trust in the authorities and the very limited (or no) costs of testing likely contributed to high compliance with testing recommendations from the government [50,51]. Hence, it is not clear whether the results of our study are generalisable also to countries with lower compliance with recommendations and trust in the health authorities.

## Conclusion

It is natural for the testing criteria and capacity to change during a pandemic. However, for disease surveillance purposes it is unfortunate when the testing regime changes towards only confirming positive cases. One possible solution would have been to include frequent, regular testing of a random (voluntary) sample of the population [49]. Testing a random sample of the population would in general allow for adjusting for different testing criteria and representability of the test data. Other alternatives would be to implement wastewater surveillance to complement the clinical surveillance [10]. The value of environmental surveillance is especially high when clinical testing is limited or the disease incidence is low [13].

## Supporting information

**S1 Text. Supporting information appendix for: Effect of testing criteria for infectious disease surveillance: The case of COVID-19 in Norway.** Additional supplementary results and methods.
(PDF)

## Author Contributions

**Conceptualization:** Solveig Engebretsen, Magne Aldrin.

**Formal analysis:** Solveig Engebretsen, Magne Aldrin.

**Investigation:** Solveig Engebretsen, Magne Aldrin.

**Methodology:** Solveig Engebretsen, Magne Aldrin.

**Visualization:** Solveig Engebretsen, Magne Aldrin.

**Writing – original draft:** Solveig Engebretsen, Magne Aldrin.

**Writing – review & editing:** Solveig Engebretsen, Magne Aldrin.

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
