## [Decision Letter · Decision Letter 0]

29 Apr 2024

PONE-D-23-42704Effect of testing criteria for infectious disease surveillance: the case of COVID-19 in NorwayPLOS ONE

Dear Dr. Engebretsen,

Thank you for submitting your manuscript to PLOS ONE. After careful consideration, we feel that it has merit but does not fully meet PLOS ONE’s publication criteria as it currently stands. Therefore, we invite you to submit a revised version of the manuscript that addresses the points raised during the review process. Please address the comments from the two reviewers. Specifically, among all critical comments, the justification of the chosen method among other alternative approaches should be carefully discussed/quantified.

We look forward to receiving your revised manuscript.

Kind regards,

Chenfeng Xiong

Academic Editor

PLOS ONE

Journal Requirements:

Reviewers' comments:

Reviewer's Responses to Questions

**Comments to the Author**

1. Is the manuscript technically sound, and do the data support the conclusions?

Reviewer #1: Yes

Reviewer #2: Yes

2. Has the statistical analysis been performed appropriately and rigorously? 

Reviewer #1: Yes

Reviewer #2: I Don't Know

3. Have the authors made all data underlying the findings in their manuscript fully available?

Reviewer #1: Yes

Reviewer #2: Yes

4. Is the manuscript presented in an intelligible fashion and written in standard English?

Reviewer #1: Yes

Reviewer #2: Yes

5. Review Comments to the Author

Reviewer #1: Review of “Effect of testing criteria for infectious disease surveillance: the case of COVID-19 in Norway”

This paper is a significant contribution which discusses how PCR testing of suspected cases varies due to testing policy criteria. To clarify its contribution, however, I recommend a major revision of the manuscript.

In the introduction, I suggest the authors provide a clearer context for this study. This could involve a more explicit comparison to what we currently understand about the role of PCR testing and its surveillance impact during COVID-19. This clarity will help readers better understand how Norway's approach compares and what unique findings this study offers. The main finding stressed by the authors is that “smaller changes in testing criteria do not seem to have large, abrupt consequences for the disease surveillance, larger changes like the introduction and massive use of self-tests makes the test data less useful for surveillance.” I think the authors need to further develop this argument in the introduction, and better explain how they arrive at this conclusion especially since the authors are only analyzing PCR testing data, even if considering changes in antigen and PCR testing policies.

I am not clear on the research design time period of the study. The main objective of the study seems to be to use PCR test data from April 1, 2020 to 2022 to evaluate how changes in antigen and PCR testing policies affected the demand for PCR testing and positivity rates when antigen tests were introduced. However, on line 111 the authors write “We fitted three generalised additive regression models (GAM) [32] to the test data for 2020.” This is similar in the abstract which states “The regression analysis focuses on the time period until 2021, i.e. before Norway started

16 vaccination.” These sentences suggest that the cutoff is December 31, 2020. In fact, this is not the case as the figures show data up until 2022. The authors should provide precise dates and the number of weeks for the start and end date of the study.

The study's objectives should be stated more clearly. The authors describe changes in PCR and antigen/lateral flow testing policies. Lateral flow tests were available in June/July 2021 and mass testing in August 2021. Testing criteria were considerably loosened in 2022. In other words, the research analyzes PCR outcomes, but changes in antigen and PCR testing policies.

Since this paper claims that its findings can be used to inform surveillance, it would be important to explain why antigen test results were used in the analysis combined with the PCR test results that were analyzed. What reporting systems were required for positivity notification for antigen and PCR specimens in Norway? How were the PCR test results obtained by the authors? Why wasn’t there research undertaken to explore laboratory capacity and scaling up over the period? Does Norway use sentinel surveillance for severe respiratory infections, and what does this data suggest compared to the overall volume and positivity rates of PCR testing in the general population?

With respect to methods, the authors could better explain why the GAM is the preferred method to analyze how testing changed over time due to changes in testing policies. Other studies have used event history analysis or structural breaks to identify how testing criteria impacted the total and positivity rates for testing, for example. As I understand this method, it has the advantage of allowing us to study an increasing growth trend considering seasonal trends. However, I think that the models as described suffer from omitted variable bias as the major policy announcements that add ripples to an otherwise smooth trend are not included in the specifications. Since these events are not included, the GAM might mistake nonlinear growth to be persistent occurrences and their effects would be erroneously propagated.

Hospitalization incidence is justified as a “proxy” for the level of “infectiousness.” I am not sure about this justification, and I think it needs to be better explained. Why don’t the authors use the lag of the dependent variable as this would also capture infectiousness?

I am not sure why the authors use the index of i and not t when they report the models that will be estimated. It would be important to explain if this is weekly or daily data and the total number of time points.

Reviewer #2: I suppose my primary concern is with the entire premise of the paper, which is that changes in test criteria and self-selection for testing of COVID-19 (similar to any other infectious disease) will result in various biases in determined incidence of infection. I would propose that this is pretty obvious and a foregone conclusion, and one of the reasons for which, in many places around the world, various researchers have implemented water-based surveillance (and other more comprehensive, less-biased methods) of attempting to measure the true incidence of SARS-CoV-2 in communities.

I would greatly appreciate if the following questions/concerns could also please be addressed:

1. What was the thought process behind choosing "time" and "weekday" as the explanatory variables in your regression model? Why not consider age, gender, socioeconomic status and other potential explanatory variables for more or less frequent testing and/or test positivity?

2. I agree that hospital incidence provides more robust surveillance data than community test data in many ways but is laden with its own issues - primarily that it is biased towards more ill, more severe cases, and not necessarily reflective of what is happening in the general population in the community. I would argue that even in the hospital setting there still dependence on testing criteria, however, and that many hospitals may still have a limited capacity depending on available resources.

3. I can certainly see that in Norway, similar to many countries, there was variable availability of different testing available for different groups over time, as well as fluctuations in restrictions. It is not surprising that the number of positive PCR tests and the denominator of total PCR tests completed also varied over time, depending on the testing protocol and COVID-19 restrictions in place.

4. How was COVID-19 hospital incidence determined? Was this based on a positive test alone, clinical criteria alone or a combination of both? This is not clearly outlined in your Materials and Methods section.

5. Your generalized additive regression models appear to be well conducted; however, I am again curious in respect to your choice of time and weekday as your only explanatory variables? Why were demographics not considered as potential explanatory variables?

6. The use of hospital incidence as a proxy for infection level in the population is reasonable (and was an approach taken in many places), however, this does have its problems and there should be some discussion around the shortcomings of this assumption. Please address this.

7. I am confused by the statement that on "May 27, 2021 Norway opened up to testing without suspicion of infection and mass testing, which seems to coincide with a decrease in both the total number of tests and proportion of positive tests." How did the denominator of total number of tests decrease if mass testing was being conducted - even without suspicion of infection?

8. I do agree with your final conclusion that regular testing of a random (voluntary) sample of the population may have been helpful in allowing for better community surveillance of COVID-19, as stated. However, personally, I believe that there are potentially better, more comprehensive and less-biased ways to measure the incidence of COVID-19 in a community locale or region (such as waste-water based surveillance, etc).

9. In conclusion, you are quite correct in that we must be clear about the purpose of our testing in any given situation--whether it is to identify cases and stop transmission or try to accurately measure incidence or prevalence of an infectious disease. I am not certain how many countries and regions have necessarily used outpatient (community) testing as a surrogate measure for the latter, as opposed to the former, especially considering changes in testing protocols and restrictions over time.

6. PLOS authors have the option to publish the peer review history of their article (what does this mean?). If published, this will include your full peer review and any attached files.

Reviewer #1: No

Reviewer #2: No

---

## [Author Response · Author response to Decision Letter 0]

12 Jun 2024

See also attached PDF, with content copied and pasted here (and note that in the PDF our response is provided in red, while tha was not possible here): 

Dear Editor, 

Thank you for the positive response to our manuscript. We are thankful for the good suggestions from the referees which have improved our manuscript. We have tried to answer the comments to the best of our ability, and we hope that we have answered the comments satisfactorily. The changes to the manuscript text are highlighted with track changes. We also provide a clean version. Below, we provide a point-by-point response to the reviewer comments, with our response in red. 

Best regards

Solveig Engebretsen and Magne Aldrin

Reviewer #1: Review of “Effect of testing criteria for infectious disease surveillance: the case of COVID-19 in Norway”

This paper is a significant contribution which discusses how PCR testing of suspected cases varies due to testing policy criteria. To clarify its contribution, however, I recommend a major revision of the manuscript.

In the introduction, I suggest the authors provide a clearer context for this study. This could involve a more explicit comparison to what we currently understand about the role of PCR testing and its surveillance impact during COVID-19. This clarity will help readers better understand how Norway's approach compares and what unique findings this study offers. The main finding stressed by the authors is that “smaller changes in testing criteria do not seem to have large, abrupt consequences for the disease surveillance, larger changes like the introduction and massive use of self-tests makes the test data less useful for surveillance.” I think the authors need to further develop this argument in the introduction, and better explain how they arrive at this conclusion especially since the authors are only analyzing PCR testing data, even if considering changes in antigen and PCR testing policies.

We have included more background on infectious disease surveillance and use of confirmed cases during the pandemic for surveillance in general and modelling in particular. Moreover, we have argued why testing criteria need to be considered in terms of the observation model used when performing inference in infectious disease models. The added paragraphs read

The different surveillance data sources have different advantages and limitations, in terms of for example accuracy, representativity, noise, timeliness, acceptance, and cost and resource demands [4, 5]. It is therefore common to combine evidence from multiple surveillance data sources. Specifically, combining evidence in different data sources was also recommended by WHO for COVID-19 surveillance purposes [6]. Another example is the routine influenza surveillance system in Norway which includes, among other sources, data on the number of general practitioner visits related to influenza, sentinel surveillance and laboratory-based surveillance of patients with acute respiratory infection, seroprevalence studies, and hospital prevalence [7]. Laboratory-based surveillance and seroprevalence studies are in general less noisy than data on influenza-like illness diagnoses from general practitioners, but more delayed and costly.

During the COVID-19 pandemic, data on confirmed cases were used for surveillance and modelling purposes in many countries, for example when estimating reproduction numbers (see for example [8, 9, 10, 11]). Specifically, confirmed cases were included in the WHO’s global COVID-19 surveillance system, in addition to data on e.g. the total number tested, deaths and hospitalisations due to COVID-19 [12]. Moreover, WHO’s guidance for public health surveillance of COVID-19 states that data on number of confirmed cases from laboratory tests should be collected as they are useful for early warning of changes in epidemiological patterns and to monitor trends in infection [6]. During the COVID-19 pandemic, multiple countries also used environmental surveillance, in particular wastewater surveillance. Such environmental surveillance has the advantage of not being affected by health care seeking behaviour, disease severity, or testing criteria/capacity [13]. Norway also used wastewater surveillance from June 2022 [10]. A systematic review on the use of environmental surveillance during COVID-19 concludes that wastewater surveillance could be used as an early warning signal and hence complement clinical surveillance of SARS-CoV-2 [14], but that the cost-benefit needs to be taken into account. Wastewater surveillance could also contribute to early detection of new variants. As noted by WHO, laboratory-confirmed cases remain the primary source of evidence on incidence of infection [13]. 

The Norwegian Institute of Public Health used both hospital incidence and confirmed cases for inference in the mathematical models of COVID-19 spread and for estimating reproduction numbers [15, 16]. Hospital incidence is a more robust surveillance source than test data, as it does not depend on testing criteria and testing capacity. Hospital incidence is also less sensitive to health seeking behaviour and self-selection. In addition, one of the main uses of mathematical modelling for real-time situational awareness is to predict hospitalisations, and hence it is important to calibrate to hospital data. There are, however, fewer observations in the hospital data than in the test data, they are more delayed, and they generally represent a population at higher risk than the community at large. Moreover, it is problematic to use hospital case data for surveillance if the hospital capacity is reached. To our knowledge, this was not a problem during the pandemic in Norway. Hence, one of the largest limitations with using hospital incidence data alone in infectious disease surveillance is that it takes more time before a change in transmission is visible in the hospital data than in the test data. It is therefore important to combine information from both data sources when calibrating mathematical disease spread models and in surveillance. Though data on confirmed cases from laboratory tests are subject to under reporting due to e.g. health care seeking behaviour, they can still be used relatively straightforward to monitor trends as long as the under reporting stays constant [5]. Nonetheless, it is difficult to use test data in periods where the testing strategy is unstable, that is, when the meaning of a positive test changes over time. If the meaning of a positive test changes, this will potentially have consequences for surveillance and estimated reproduction numbers [17]. Changes in testing criteria may lead to conflicting information in different data sources used for surveillance and/or for inference in mathematical models. If the changes in testing criteria have a practically significant effect on the outcomes of the models, then a complex observation model for the test data would be necessary. This is a nontrivial inference problem and will likely be subject to identifiability issues in a complex observation model unless the parameters are assumed fixed or given informative priors. In Norway, the modelling team from the Norwegian Institute of Public Health handled this in part by incorporating the total number of tests in a relatively simple observation model, as a proxy for changes in testing criteria and testing capacity. The test data were not used for model inference in the early period when testing had not stabilised. Moreover, a seven days rolling mean of the test data was used in the calibration, in order to take into account potential day-of-the-week effects [15]. Using the number of tests to estimate the reproduction number for COVID-19 was also done in for example [18, 19], and another approach to an observation model which takes into account changes in testing criteria can be found in [20].

I am not clear on the research design time period of the study. The main objective of the study seems to be to use PCR test data from April 1, 2020 to 2022 to evaluate how changes in antigen and PCR testing policies affected the demand for PCR testing and positivity rates when antigen tests were introduced. However, on line 111 the authors write “We fitted three generalised additive regression models (GAM) [32] to the test data for 2020.” This is similar in the abstract which states “The regression analysis focuses on the time period until 2021, i.e. before Norway started

16 vaccination.” These sentences suggest that the cutoff is December 31, 2020. In fact, this is not the case as the figures show data up until 2022. The authors should provide precise dates and the number of weeks for the start and end date of the study.

Under the «Methods» section we now specify further the time periods used in the different parts of the paper. Specifically, we have added the following paragraph: 

We analysed the data using two different approaches. The first was a purely explorative data analysis, where we inspected the data from the whole period available, that is, from April 1, 2020 to November 13, 2022. In addition, we fitted regression models to the daily test data from April 1, 2020 to January 1, 2021 (in total 274 days, one day missing). We chose to limit the regression analysis to this period only, as it was not cluttered by larger known changes in variant and by vaccination.

The study's objectives should be stated more clearly. The authors describe changes in PCR and antigen/lateral flow testing policies. Lateral flow tests were available in June/July 2021 and mass testing in August 2021. Testing criteria were considerably loosened in 2022. In other words, the research analyzes PCR outcomes, but changes in antigen and PCR testing policies.

Since this paper claims that its findings can be used to inform surveillance, it would be important to explain why antigen test results were used in the analysis combined with the PCR test results that were analyzed. What reporting systems were required for positivity notification for antigen and PCR specimens in Norway? How were the PCR test results obtained by the authors? Why wasn’t there research undertaken to explore laboratory capacity and scaling up over the period? Does Norway use sentinel surveillance for severe respiratory infections, and what does this data suggest compared to the overall volume and positivity rates of PCR testing in the general population?

In our study, we only use the publicly available number of confirmed cases made available from the Norwegian Institute of Public Health. These results were obtained by downloading them from the github data repository, as explained under the Data section. We do not have access to data on the number and results of self-tests, and such data were not systematically collected in Norway. As we do not use the self-tests in this study and only the PCR tests, we believe that it is not within the scope of our study to go into details on the reporting systems for self-tests in Norway. However, they were never systematically reported and recorded, particularly not the negative self-tests. We have added this information in the manuscript. Moreover, we have included more information in general on surveillance of infectious diseases, and we also briefly mention sentinel surveillance for SARI/influenza. However, we do not have access to these data and hence cannot compare them to the COVID-19 test data. The added paragraph on surveillance in general (also repeated in point 1) reads:

The different surveillance data sources have different advantages and limitations, in terms of for example accuracy, representativity, noise, timeliness, acceptance, and cost and resource demands [4, 5]. It is therefore common to combine evidence from multiple surveillance data sources. Specifically, combining evidence in different data sources was also recommended by WHO for COVID-19 surveillance purposes [6]. Another example is the routine influenza surveillance system in Norway which includes, among other sources, data on the number of general practitioner visits related to influenza, sentinel surveillance and laboratory-based surveillance of patients with acute respiratory infection, seroprevalence studies, and hospital prevalence [7]. Laboratory-based surveillance and seroprevalence studies are in general less noisy than data on influenza-like illness diagnoses from general practitioners, but more delayed and costly.

With respect to methods, the authors could better explain why the GAM is the preferred method to analyze how testing changed over time due to changes in testing policies. Other studies have used event history analysis or structural breaks to identify how testing criteria impacted the total and positivity rates for testing, for example. As I understand this method, it has the advantage of allowing us to study an increasing growth trend considering seasonal trends. However, I think that the models as described suffer from omitted variable bias as the major policy announcements that add ripples to an otherwise smooth trend are not included in the specifications. Since these events are not included, the GAM might mistake nonlinear growth to be persistent occurrences and their effects would be erroneously propagated.

We have tried different regression models to study the effect over time. It is difficult to interpret the results from models with structural breaks (regression models where we allow for a change in level on different dates related to changes in testing policies) as we cannot separate the effect of changes on specific days from gradual changes over time due to other, unknown factors. Moreover, the effect of changes in testing criteria are likely gradual, and similarly changes in testing capacity are also likely gradual. We have therefore concluded that a smoothing spline in time was most appropriate for our study. Major policy announcements would be captured in the smoothing spline, though they might be smoothed. The smoothing parameter is selected through the in-built default smoothing parameter selection in the R-package mgcv. The general experience with this package is that the resulting effects are too volatile, i.e. not smooth enough, though this would of course not be true for an underlying truth containing discrete jumps. We have added a comment on the fact that we have also tried models with sudden jumps in the Methods section: 

Note that we also tried out models which allowed for sudden jumps in accordance with the changes in testing criteria/policy. However, it turned out that the changes in number of tests and positive tests in 2020 were gradual and did not occur as sudden jumps.

Hospitalization incidence is justified as a “proxy” for the level of “infectiousness.” I am not sure about this justification, and I think it needs to be better explained. Why don’t the authors use the lag of the dependent variable as this would also capture infectiousness?

One of the key objectives of the study is to analyse the changes in testing criteria/capacity over time. This effect should ideally be adjusted for the infection level. We therefore adjust for the hospital incidence. We cannot include the lag of the dependent variable to capture infectiousness through for example an autoregressive model, as this would not allow us to study changes over time. This approach could, however, have been used if we were only interested in the weekday effects. 

We have added the following to the discussion: 

We adjusted for the hospital incidence as a proxy for the infection level in the population. This is unproblematic as long as the hospital incidence is a constant proportion of the number of infected individuals. However, this would not be the case if there were changes in viral variants/properties such that the severity of infected cases changed. If the hospital capacity is reached, then the hospital incidence data need to be treated as censored data, and that would have to be taken into account in the analysis. To our knowledge, the hospital capacity was not reached in Norway during the pandemic. 

I am not sure why the authors use the index of i and not t when they report the models that will be estimated. It would be important to explain if this is weekly or daily data and the total 

---

## [Decision Letter · Decision Letter 1]

5 Aug 2024

Effect of testing criteria for infectious disease surveillance: the case of COVID-19 in Norway

PONE-D-23-42704R1

Dear Dr. Engebretsen,

We’re pleased to inform you that your manuscript has been judged scientifically suitable for publication and will be formally accepted for publication once it meets all outstanding technical requirements.

Kind regards,

Chenfeng Xiong

Academic Editor

PLOS ONE

Additional Editor Comments (optional):

Reviewers' comments:

Reviewer's Responses to Questions

**Comments to the Author**

1. If the authors have adequately addressed your comments raised in a previous round of review and you feel that this manuscript is now acceptable for publication, you may indicate that here to bypass the “Comments to the Author” section, enter your conflict of interest statement in the “Confidential to Editor” section, and submit your "Accept" recommendation.

Reviewer #1: All comments have been addressed

Reviewer #2: All comments have been addressed

2. Is the manuscript technically sound, and do the data support the conclusions?

Reviewer #1: Yes

Reviewer #2: Yes

3. Has the statistical analysis been performed appropriately and rigorously? 

Reviewer #1: Yes

Reviewer #2: Yes

4. Have the authors made all data underlying the findings in their manuscript fully available?

Reviewer #1: Yes

Reviewer #2: Yes

5. Is the manuscript presented in an intelligible fashion and written in standard English?

Reviewer #1: Yes

Reviewer #2: Yes

6. Review Comments to the Author

Reviewer #1: (No Response)

Reviewer #2: You have answered all of my questions and concerns to my satisfaction. I believe your manuscript to be a valuable contribution to the peer-reviewed literature.

7. PLOS authors have the option to publish the peer review history of their article (what does this mean?). If published, this will include your full peer review and any attached files.

Reviewer #1: No

Reviewer #2: No

---

## [Editor Report · Acceptance letter]

7 Aug 2024

PONE-D-23-42704R1 

PLOS ONE

Dear Dr. Engebretsen, 

I'm pleased to inform you that your manuscript has been deemed suitable for publication in PLOS ONE. Congratulations! Your manuscript is now being handed over to our production team.

Kind regards, 

on behalf of

Dr. Chenfeng Xiong 

Academic Editor

PLOS ONE